# Ribonucleotide reductase regulatory subunit M2 (RRM2) as a potential sero-diagnostic biomarker in non-small cell lung cancer

**Dandan Zhou**[1,2], **Xiuming Zhai**[1]*, **Ruixue Zhang**[3]

**1** Department of Clinical Laboratory, Chongqing Hospital of Traditional Chinese Medicine, Chongqing, 400021, China, **2** Chongqing Key Laboratory of Sichuan-Chongqing Co-Construction for Diagnosis and Treatment of Infectious Diseases Integrated Traditional Chinese and Western Medicine, Chongqing, 400021, China, **3** School of Laboratory Medicine, Chongqing Medical University, Chongqing, 400016, China

* xiumingzhai@163.com

## Abstract

### Objectives

Non-small cell lung cancer (NSCLC) is a major cause of cancer-related death worldwide. Most cases are diagnosed at an advanced stage using current tumor markers. Here, we aimed to identify potential novel potential biomarkers for NSCLC.

### Material/Methods

Four independent datasets from the Gene Expression Omnibus database were analyzed. The relative expression of ribonucleotide reductase regulatory subunit M2 (RRM2) mRNA in 30 paired of NSCLC paired tissues was measured by reverse transcription quantitative PCR. Serum levels of cytokeratin fragment 21–1 (CYFRA21-1), pro-gastrin-releasing pep-tide (ProGRP), carcinoembryonic antigen (CEA), and neuron-specific enolase (NSE) were measured using electrochemiluminescence immunoassays, and serum RRM2 levels were evaluated by an enzyme-linked immunosorbent assay.

### Results

The mRNA expression level of RRM2 was significantly increased in most NSCLC lesions compared to para-adjacent tissues. Serum RRM2 levels in NSCLC patients were significantly elevated compared to healthy controls and were also associated with distant metastasis and histological type, but not with tumor size or lymph node metastasis. Receiver operating characteristic curve analysis showed a higher diagnostic ratio for NSCLC using RRM2 alone compared to other traditional tumor markers.

### Conclusions

RRM2 is a potential sero-diagnostic biomarker for NSCLC.

**Data Availability Statement:** All relevant data are within the paper and its Supporting Information files.

**Funding:** This work was financially supported by the Xinglin Scholar Research Premotion Project of

Chengdu University of TCM (Grant No.:
YYZX2020069). The grant recipient is Zhou, D.D.,
and the funder provided assistance in study design,
data collection, and manuscript analysis.

## Introduction

Lung cancer has become a leading cause of malignant tumor-related deaths worldwide, and in China, it has the highest mortality rate of cancer cases [1]. Lung cancer is broadly classified into small cell lung cancer (SCLC) and non-small cell lung cancer (NSCLC). NSCLC is the most common subtype of lung cancer subtype (approximately 85% of all cases) [2], and includes lung squamous cell carcinoma (LUSC), lung adenocarcinoma (LUAD),large cell carcinoma (LCC), and other less frequently-diagnosed histological types [3]. However, due to metastasis spread, NSCLC tumors are highly aggressive, and approximately 80% of patients are diagnosed at an advanced stage, often leading to treatment failure and death. Although the efficacy of current treatments for NSCLC has greatly improved, the 5-year survival rate for these patients is still less than 18%, partly because most cases are diagnosed at an advanced stage [4].

Currently, the diagnosis of lung cancer relies on spiral computed tomography and X-rays; however, it is prone to both false-negative and false-positive cases [5]. Serum biomarkers have been identified and used as critical tools in the diagnosis of NSCLC. Unfortunately, traditional tumor markers, such as cytokeratin fragment 21–1 (CYFRA21-1), pro-gastrin-releasing peptide (ProGRP), carcinoembryonic antigen (CEA), and neuron-specific enolase (NSE), are limited in clinical use due to their poor sensitivity and specificity values [6–8]. Notably, several novel biomarkers for the diagnosis of NSCLC have been identified in recent years due to their higher sensitivity and specificity compared to traditional tumor markers, including exosomes [9,10], circulating tumor DNA (ctDNA) [11,12], microRNAs [13], CircRNAs [14], and long non-coding RNA [15,16]. However, there are still many challenges associated with their widespread use in population screening, including restrictive detection technology, equipment complexity, and high testing costs. Therefore, it is imperative to identify early diagnostic biomarkers to increase the diagnostic rate, which may contribute to improving the therapeutic status and prognosis of NSCLC patients. With the rapid development of high-throughput technology, tumor genome sequences can be obtained to analyze all relevant tumor genome copy numbers, gene correlations, or polymorphisms [17,18]. This provides the possibility of using bioinformatics to mine cancer-related gene expression profiles.

Ribonucleotide reductase regulatory subunit M2 (RRM2) encodes one of the two distinct subunits of ribonucleotide reductase that catalyzes the conversion of ribonucleotides to deoxyribonucleotides [19]. Several studies have reported increased expression of RRM2 mRNA in NSCLC tissues [20,21] and cell lines [16,21]. However, it is unclear whether RRM2 can be used as a diagnostic marker for NSCLC. RRM2 protein is a secreted protein and can be detected in serum. In this study, we aim to investigate the possibility of RRM2 as a serum diagnostic marker for NSCLC based on bioinformatics.

## Materials and methods

### Data resources

Series matrix files of GSE18842, GSE19188, GSE30219, and GSE40791 were downloaded from the Gene Expression Omnibus (GEO) database. All platforms were based on GPL570 (Affymetrix Human Genome U133 Plus 2.0 Array). A total of 224 normal samples and 380 NSCLC samples were included in the study. Of these, 45 normal and 46 tumor samples were obtained from GSE18842, 65 normal and 91 tumor samples from GSE19188, 14 normal and 149 tumor samples from GSE30219, and 100 normal and 94 tumor samples from GSE40791. All matrix files data are available online.

The NSCLC group included NSCLC patients treated at the Chongqing Traditional Chinese Medicine Hospital from October 2020 to July 2022. During the data collection process, we were able to simultaneously identify the information about the medical history of individual participants. The TNM staging of NSCLC was based on the 8th edition of the AJCC TNM staging system [22]. The exclusion criteria were as follows: 1) patients with incomplete medical records; 2) patients without a clear histopathological or cytological diagnosis; 3) patients with tumors of other systems; 4) patients with severe cerebrovascular and cardiovascular diseases, diabetes, or autoimmune diseases; and 5) patients who received radiotherapy, chemotherapy, or immunotherapy. A total of 30 tissue samples were collected from patients with NSCLC, and serum samples were collected from 110 patients with NSCLC. In addition, 50 patients who underwent physical examination at our hospital during the same period were selected as the healthy control group. The study was approved by the Ethics Committee of Chongqing Traditional Chinese Medicine Hospital, and written informed consent was obtained from all patients.

### Screening for differentially expressed genes (DEGs)

Bioinformatic analysis was performed using R software (v3.6.2; http://www.r-project.org). The GEOquery, limma, and dplyr packages were used for data normalization, and the RobustRankAggreg package was used to screen the DEGs using an adjusted $P$-value < 0.05 and |log2 fold change (FC)| $\geq$ 2 as the cut-off criteria. The VennDiagram R package was used to identify significant co-expressed genes.

### Gene ontology (GO) and Kyoto Encyclopedia of Genes and Genomes (KEGG) pathway enrichment analysis of DEGs

GO [23], a tool for annotating genes from various ontologies using standard expression terms, currently includes three aspects of biological content: cellular components (CC), biological processes (BP), and molecular functions (MF). KEGG is a bioinformatics resource for "understanding the advanced functions and utility resource library of biological systems", which mainly represents intermolecular interactions and intermolecular networks [24]. The DAVID database (v6.8; http://david.abcc.ncifcrf.gov), an online tool, was utilized to perform GO enrichment and KEGG pathway analysis, using "after FDR" (corrected $P$-value < 0.05) for statistical significance. The visualized results of GO functional enrichment were presented using the ggplot2 package in R.

### Protein-protein interaction (PPI) network and hub gene analysis

PPI network of DEGs was obtained using the online analysis website STRING (v11.0; https://string-db.org/). Subsequently, visual analysis of the PPI network graphs was performed using Cytoscape (v3.7.2; https://cytoscape.org) and hub genes were analyzed using the Cytoscape plugin CytoHubba. The Matthews Correlation Coefficient (MCC) algorithm was used to identify the top ten hub genes. Hub genes were subjected to KEGG pathway analysis using the DAVID database.

### Survival analysis and validation of hub genes

Gene Expression Profiling Interactive Analysis (GEPIA) is a web-based tool that integrates The Cancer Genome Atlas (TCGA) and Genotype-Tissue Expression(GTEx) data for cancer and normal gene expression profiling and interactive analysis, including differential expression analysis, patient survival analysis, profiling plotting, correlation analysis, similar gene

detection, and dimensionality reduction analysis. To further validate the 10 hub genes, the expression levels were analyzed and Kaplan-Meier (KM) survival curves were mapped using GEPIA (http://gepia.cancer-pku.cn/index.html).

## The relevance of hub genes to immune activity in the tumour microenvironment (TME)

First, the single sample gene set enrichment analysis (ssGSEA) algorithm was used to normalize the gene expression values of the NSCLC samples. Then, we applied the empirical cumulative distribution function [25] to calculate the enrichment fraction of 29 immune cell types [26], which facilitated the categorization of NSCLC samples into distinct immune activity sets. Subsequently, we employed the GSVA software package to stratify patients into high and low immune activity groups. To validate the accuracy of immune activity grouping based on the GSE19188 data, we utilized the ESTIMATE method [27], which reflects the infiltration levels of immune cells and stromal cells within the TME, based on their specific gene expression levels.

## Laboratory analysis

**Tissue RNA assays.** The RRM2 mRNA expression was detected using reverse transcription quantitative PCR (RT-qPCR) in 18 pairs of LUSC tissues and 12 pairs of LUAD tissues. Total RNA was isolated from NSCLC patient tissue samples using TRIzol reagent (Invitrogen, Carlsbad, CA, USA). Quantity of RNA was determined using a NanoDrop ND-1000 spectrophotometer (Thermo Fisher Scientific, Waltham, MA, USA). Reverse transcription of RNA into cDNA was performed according to the instructions of the Takara kit (Takara Bio Inc., Japan). The SYBR Green PCR Master Mix System (Tiangen Biotech, Beijing, China) was used for RT-qPCR reactions. GAPDH was used as a control for comparison of relative expression of RRM2 mRNA. In the RT-qPCR experiment, three replicate wells were performed. The primer sequences are shown in Table 1, and the primers were supplied by Sangon Biotech (Shanghai) Co., Ltd.

**Serum protein assays.** Prior to treatment, approximately 5 mL of peripheral blood samples were collected from the test subjects in sterile tubes without anticoagulants. All samples were centrifuged at 4000 rpm for 10 minutes at room temperature. Finally, the serum samples were stored at -20°C until analysis. Serum RRM2 levels were measured using an enzyme-linked immunosorbent assay (ELISA) on a TECAN Freedom EVOlyzer-2 150 platform (Tecan, Männedorf, Switzerland). The RRM2 ELISA kit was purchased from Shanghai Tongwei Biotechnology Co. Ltd. (Shanghai, China). According to the manufacturer's instructions, the RRM2 absorbance was measured at 450 nm, and RRM2 concentrations were calculated using an appropriate calibration curve. RRM2 results are shown in ng/mL. Serum CEA, CYFRA21-1, ProGRP, and NSE levels were analyzed by electrochemiluminescence immunoassay (ECLIA) on a Cobas 6000 e601 instrument (Roche Diagnostics, Mannheim, Germany) with the same batch of reagents. The normal reference intervals were less than 5 ng/mL for

**Table 1. Primers sequence of target gene and internal reference gene.**

| Gene | Primers |
| --- | --- |
| RRM2 | forward primer: 5'- GCGATTTAGCCAAGAAGTTCAGAT-3' |
| | reverse primer: 5'- CCCAGTCTGCCTTCTTCTTGA-3' |
| GAPDH | forward primer:5'- AGGTCGGTGTGAACGGATTTG-3' |
| | reverse primer:5'-GGGGTCGTTGATGGCAACA-3' |

CEA, less than 3.3 ng/mL for CYFRA21-1, less than 16.3 ng/mL for NSE, and ranged from 25.3 to 69.2 pg/mL for ProGRP.

## Statistical analysis

Statistical analyses were performed using SPSS software (version 21.0; IBM Corp., Armonk, NY, USA). RT-qPCR results were expressed as $\triangle Ct$ ($\triangle Ct = Ct^{RRM2} - Ct^{GAPDH}$), and differences between the two datasets were assessed by unpaired $t$-test. The chi-square test was used to analyze the count data. The measurement data was first examined for normal distribution. Normal distribution data are displayed as the mean ± standard deviation (SD), and comparisons between two groups were performed using an unpaired $t$-test. Data that were not normally distributed are displayed as median means (quartile) [M ($P25$, $P75$)], and the Mann–Whitney U-test was used for comparison between two groups. The Kruskal-Wallis H-test was used for comparisons between multiple groups. Correlations between indicators were analyzed using Spearman's correlation analysis. Logistic regression analysis and receiver operating characteristic (ROC) curves were used to assess the diagnostic capability of biomarkers in NSCLC patients. All tests were considered statistically significant at $P$-value $< 0.05$.

## Results

### DEGs in NSCLC and normal tissues

In this study, four series of matrix files (GSE18842, GSE19188, GSE30219, GSE40791) that included 380 NSCLC and 224 normal samples overall, were used to identify DEGs ($P < 0.05$, | logFC| $\geq 2$). Venn diagrams showed that 258 genes were co-expressed, of which 91 were clearly

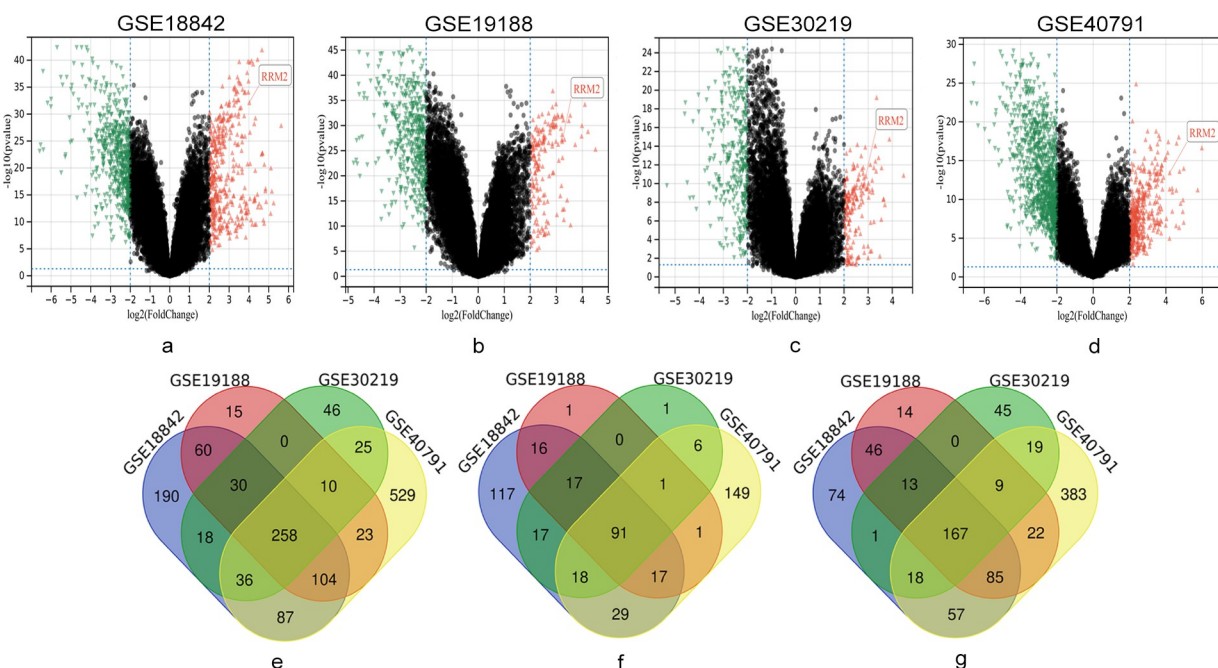

**Fig 1. Data from GSE18842, GSE19188, GSE30219 and GSE40791 were used to identify DEGs between NSCLC and normal samples.** The volcano plots show all the expressed genes from GSE18842 (a), GSE19188 (b), GSE30219 (c), GSE40791 (d). Red color represents up-regulated genes, mediumaquamarine color represents down-regulated genes. A total of 258 co-expressed genes were screened (e), of which 91 genes were significantly up-regulated (g) and 167 genes were significantly down-regulated (f).

up-regulated and 167 were clearly down-regulated (**Fig 1**). The detailed results are provided in **S1 File**.

## GO and KEGG pathway enrichment analysis

We further investigated the biological functions of the 258 DEGs. GO enrichment and KEGG pathway analysis were performed using the DAVID database. A total of 48 significantly related biological processes and 5 KEGG pathways were obtained from the up-regulated DEGs. GO analysis revealed that up-regulated DEGs showed enrichment of molecular function in micro-tubule binding, microtubule motor activity, ATP binding and protein serine/threonine kinase activity (**Fig 2A**). KEGG analysis showed that up-regulated DEGs tended to be enriched in cell cycle, progesterone-mediated oocyte maturation, p53 signaling pathway, oocyte meiosis, human T-cell leukemia virus 1 infection. However, a total of only 15 significantly related biological processes and 0 KEGG pathways were obtained from the down-regulated DEGs. GO analysis showed that down-regulated DEGs revealed enrichment of biological processes in complement activation (**Fig 2B**).

## PPI network and module analysis

We further examined which of the 258 DEGs are most likely to be central regulatory genes for NSCLC. PPI network was constructed using the STRING database and Cytoscape software. Downregulated genes are indicated by arrows and the upregulated genes are represented by circles. The biological content and pathways in which the gene is involved are represented in different colors (**Fig 3**). Subsequently, according to the MCC algorithm, the top 10 hub genes were screened through the cytoHubba, including RRM2, CDK1, UBE2C, MAD2L1, BUB1B, CCNA2, KIF20A, BUB1, KIF11, and CCNB2. We found that the 10 hub genes were significantly upregulated.

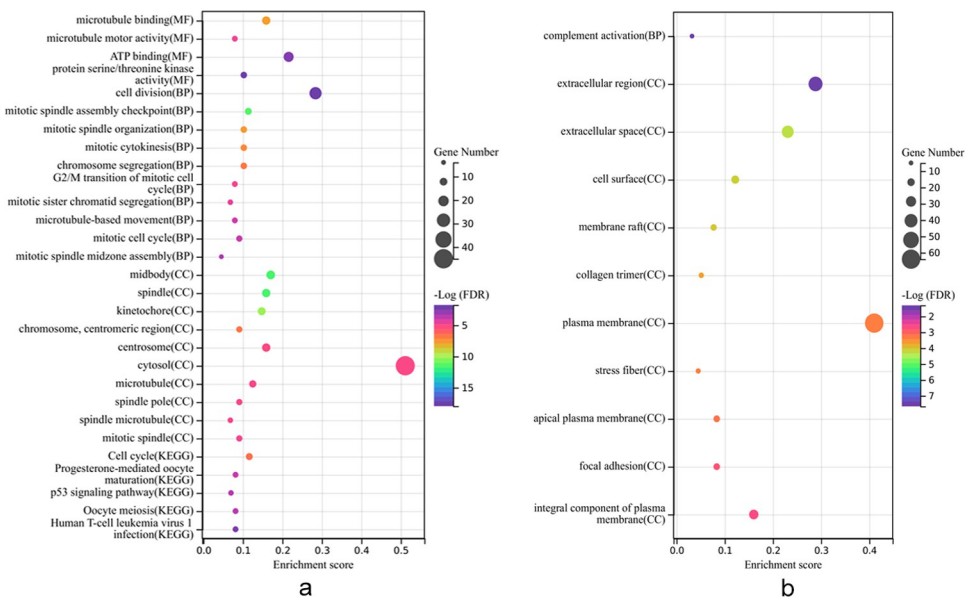

**Fig 2. The biological function of the 258 differentially expressed genes (DEGs).** GO and KEGG enrichment analysis of up-regulated DEGs (a) and down-regulated DEGs (b).

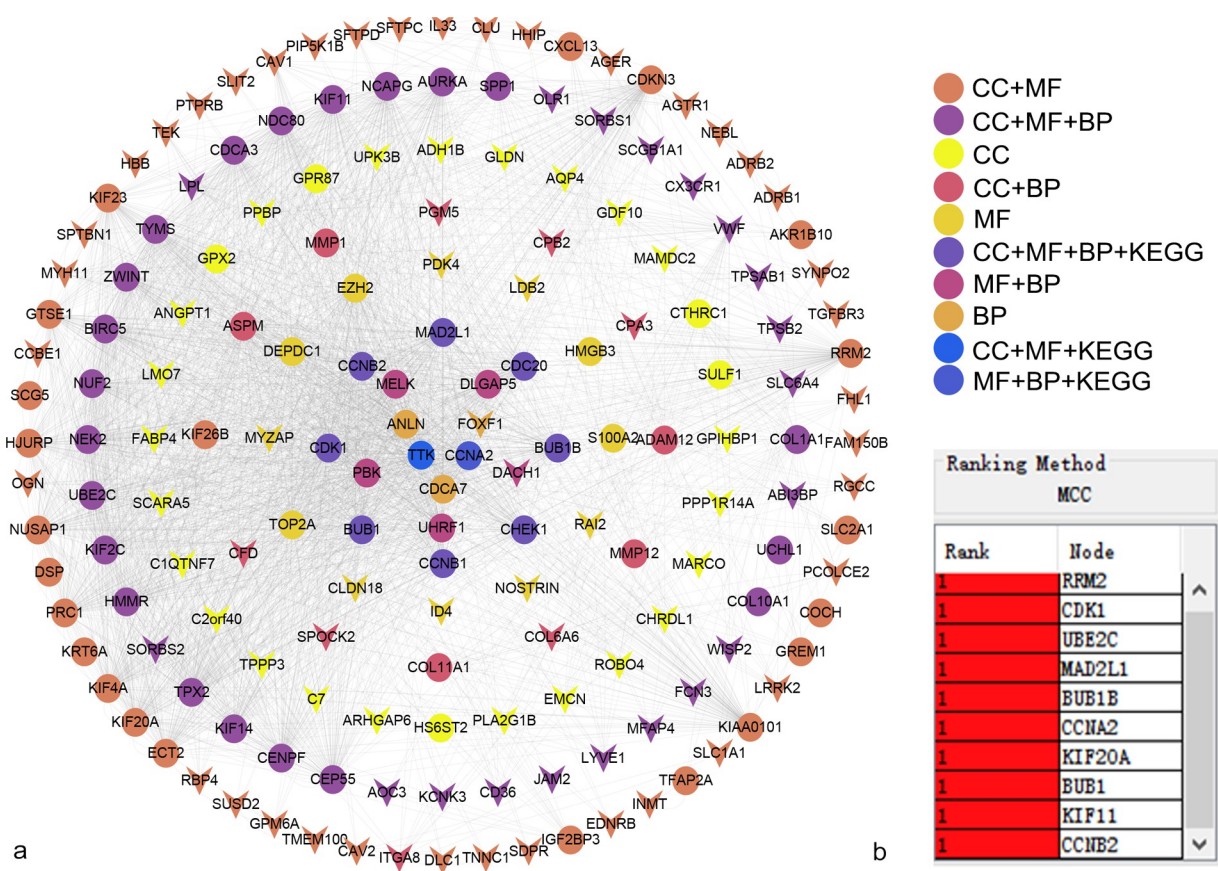

**Fig 3. Protein-protein interaction (PPI) network of the differentially expressed genes (DEGs).** (a) The arrows represent downregulated genes, while the circles represent upregulated genes. Different colors indicate that the gene is involved in different biological content and pathways and (b) Identification of the top 10 hub DEGs using the Matthews correlation coefficient (MCC) algorithm.

## Validation and survival analysis of hub genes

The expression levels of the 10 hub genes in NSCLC and normal tissues were further verified using the GEPIA database (483 LUAD vs. 347 normal, 486 LUSC vs. 338 normal). The expression levels of these hub genes were increased in both LUSC and LUAD, among which RRM2, UBE2C, MAD2L1, KIF20A, CDK1, and CCNB2 were statistically significant ($P < 0.05$); KIF11, CCNA2, BUB1B, and BUB1 were only statistically significant in LUSC ($P < 0.05$) but not in LUAD ($P > 0.05$). Based on the available reports, we found that the mRNA and protein levels of RRM2 [28], UBE2C [29], and CDK1 [30] were significantly upregulated in the large cell lung cancer cell line NCI-H460, whereas the expression of MAD2L1, KIF20A, and CCNB2 in the large cell lung cancer cell line NCI-H460 has not been reported. In addition, overall survival analysis and disease-free survival (RFS) of the 10 hub genes were performed using the GEPIA database. KM analysis showed that the survival rates of NSCLC patients with high expression levels of RRM2 or KIF11 were statistically significantly lower ($P < 0.05$). However, the survival rates of those with high expression levels of UBE2C, MAD2L1, KIF20A, CDK1, CCNB2, CCNA2, BUB1B, and BUB1 were not significantly different ($P > 0.05$). RFS results showed that high expression of RRM2, KIF20A, CDK1, CCNB2, CCNA2, BUB1B and BUB1 in LUAD patients were associated with shorter survival in LUAD patients ($P < 0.05$), whereas expression levels of UBE2C, MAD2L1 and KIF11 were not associated with survival in LUAD patients ($P > 0.05$). RFS in LUSC patients was not associated with the expression levels of any

of the 10 hub genes ($P > 0.05$). The expression levels and survival analysis of the 10 hub genes are shown in the supplemental materials (S2 File). From the above analysis, we found that RRM2 is a highly significant hub gene in NSCLC and may play an important role in NSCLC prognosis.

## Patient sample analysis

From our institution, 110 NSCLC patients were included in the NSCLC group. In addition, 50 individuals were selected as the healthy control group. There were no statistically significant differences in the basic characteristics (sex and age) between the NSCLC and healthy control groups (Age: $P = 0.462$; Gender: $P = 0.112$).

**Tissue analysis.** The RRM2 mRNA expression was detected using RT-qPCR in 18 pairs of LUSC tissues and 12 pairs of LUAD tissues. Fig 4 shows that the expression levels of RRM2 were significantly higher in LUSC tissues ($P<0.001$) and LUAD tissues ($P<0.05$) than in adjacent tissues. Combined with the above analysis, we tentatively concluded that RRM2 may be a potential biomarker to NSCLC diagnosis.

**Serum analysis.** The protein encoded by RRM2 is a secreted protein that can be detected in serum. The RRM2 serum levels in NSCLC patients and healthy controls were further verified using ELISAs. Simultaneously, serum CEA, CYFRA21-1, NSE, and ProGRP served as control markers. Serum levels of CEA, CYFRA21-1, NSE, and ProGRP were analyzed using

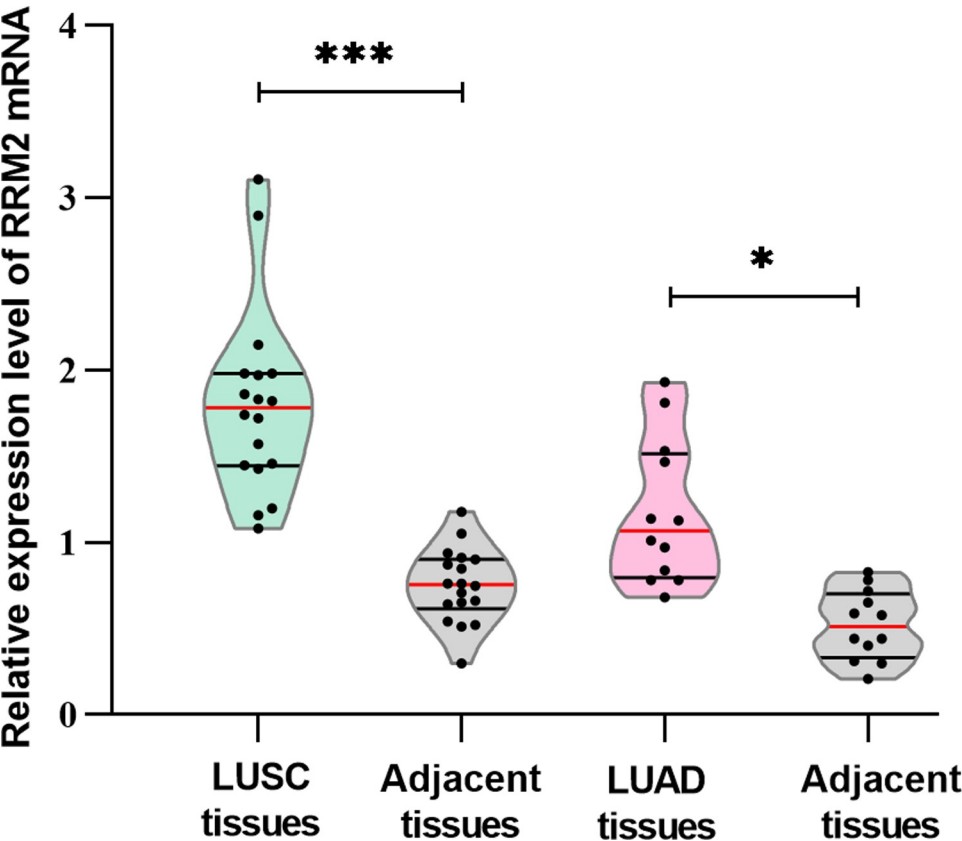

**Fig 4. The relative expression level of RRM2 mRNA in 30 pairs of NSCLC paired tissues.** The expression levels of RRM2 were significantly higher in LUSC tissues and LUAD tissues than adjacent tissues.*: $P<0.05$, **: $P<0.01$, ***: $P<0.001$.

ECLIAs. RRM2 secretion levels were significantly higher in NSCLC patients compared to healthy controls (Table 2).

Next, we analyzed the association between serum RRM2 levels and clinical characteristics of the patients. We found that the serum RRM2 level of NSCLC patients were related to distant metastasis [Yes: 158.68 (145.03, 169.30) VS. No: 139.80 (113.61, 156.87), $P = 0.001$] or histological type [Adenocarcinoma: 154.93 (136.11, 167.24) VS. Squamous cell carcinoma: 137.86 (120.40, 161.07), $P = 0.043$], but not to lymph node metastasis [Yes: 152.18(135.82, 166.72) VS. No: 144.21(99.35, 167.00), $P = 0.201$] or tumor size (Diameter $\leq$ 3 cm VS. 3 cm $<$ Diameter $\leq$ 5cm VS. 5 cm $<$ Diameter $\leq$ 7cm VS. 7 cm $<$ Diameter, $P = 0.096$). Notably, the relationship between serum RRM2 levels and TNM stage in NSCLC patients was only statistically significant between stages II and IV ($P = 0.009$), but not between the other groups ($P > 0.05$).

Then, ROC curve analysis was used to calculate the sensitivity and specificity values of the five biomarkers in NSCLC patients. The ROC analysis in Table 3 shows that the ability to distinguish NSCLC patients from healthy controls using RRM2 was 0.798 (95% confidence interval (CI) 0.731–0.866) and the cut-off value was 131.31 ng/mL. The sensitivity and specificity values were 73.6% and 74.0%, respectively. The ROC curve analysis results showed that the diagnosis of NSCLC with CEA (area under the ROC curve (AUC): 0.903, cut-off value: 2.10 ng/mL, sensitivity: 0.845, specificity: 0.880, accuracy: 0.856) was better than that with RRM2 (AUC: 0.798, accuracy: 0.738). However, when the cut-off value of CEA is adjusted according to the instructions (cut-off value: 5.0 ng/mL, sensitivity: 0.500, specificity: 0.980, accuracy: 0.650), the diagnosis of NSCLC with CEA (accuracy: 0.650) is much lower than that with RRM2 (accuracy: 0.738). Additionally, the cut-off values of CYFRA21-1, NSE, and PROGRP analyzed by ROC curve analysis are consistent with the cut-off values of each other's reagent instructions. Therefore, CEA data should be excluded from this study, partly because there may be so few subjects included that the data have a skewed distribution. From the above analysis, we believe that the single use of RRM2 has a higher diagnostic yield for NSCLC compared to the single use of traditional tumour markers (CYFRA21-1, NSE and ProGRP).

Finally, logistic regression analysis revealed that NSCLC was closely associated with RRM2 and NSE levels. Logistic regression analysis was used to combine the three biomarkers for further analysis. The AUC value of the combined score (AUC = 0.868, sensitivity = 70.9%, specificity = 90.0%) indicated that the combined detection of RRM2 and NSE had the highest efficacy in the diagnosing NSCLC (**Fig 5**).

## The relationship between RRM2 gene and immune activity in TME

In addition, we further assessed the relationship between RRM2 and immune activity to elucidate the mechanism by which RRM2 promotes the NSCLC development. Firstly, we evaluated the GSE19188 expression matrix using the ESTIMATE algorithm to obtain the immune cell score, stromal cell score, ESTIMATE score (total score), and tumor purity score in TME (Fig

**Table 2. Tumor marker levels in peripheral blood of NSCLC patients andhealthy controls.**

| Indicators | NSCLC (n = 110) | Healthy control (n = 50) | Z value | P value |
|---|---|---|---|---|
| RRM2 (ng/ml) | 151.72(129.37,166.98) | 111.78(90.17,132.65) | -6.043 | < 0.001 |
| CEA (ng/ml) | 5.00(2.85,12.39) | 1.36(0.86,1.95) | -8.165 | < 0.001 |
| CYFRA21-1 (ng/ml) | 2.97(1.92,5.19) | 1.82(1.39,2.25) | -5.279 | < 0.001 |
| NSE (ng/ml) | 19.23(16.27,26.14) | 15.17(13.19,16.65) | -5.586 | < 0.001 |
| PROGRP (pg/ml) | 48.48(37.27,65.60) | 37.50(32.48,47.90) | -3.326 | 0.001 |

**Table 3. Receiver operating characteristic (ROC) curve analysis results of indicators as biomarkers for non-small cell lung cancer (NSCLC).**

| Indicators | AUC | SE | P | 95% C.I. | Cut off | Sensitivity | Specificity | Accuracy |
|---|---|---|---|---|---|---|---|---|
| CEA | 0.903 | 0.024 | <0.001 | 0.857–0.950 | 2.10 | 0.845 | 0.880 | 0.856 |
| PROGRP | 0.664 | 0.045 | 0.001 | 0.577–0.752 | 68.99 | 0.209 | 0.940 | 0.438 |
| CYFRA21-1 | 0.761 | 0.037 | <0.001 | 0.688–0.834 | 3.31 | 0.445 | 0.940 | 0.600 |
| NSE | 0.776 | 0.037 | <0.001 | 0.703–0.849 | 16.25 | 0.755 | 0.720 | 0.738 |
| RRM2 | 0.798 | 0.034 | <0.001 | 0.731–0.866 | 131.31 | 0.736 | 0.740 | 0.738 |

6A). Further analysis demonstrated that the ESTIMATE score, immune cell score, and stromal cell score were positively correlated with immune activity (Fig 6B-6D), while tumor purity showed an inverse correlation with immune activity (Fig 6E). We ultimately found no correlation between RRM2 expression and immune activity ($P > 0.05$) in Fig 6F.

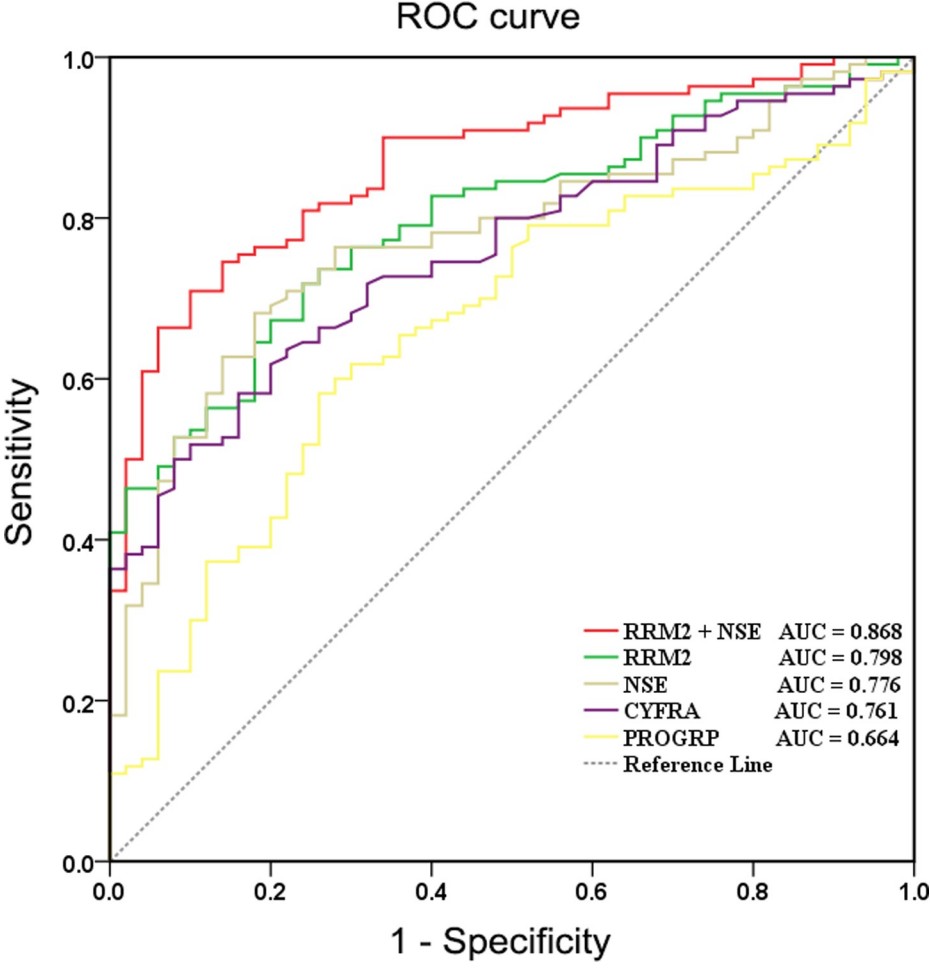

**Fig 5. Receiver operating characteristic (ROC) curves for various markers.** ROC curves showed the combined detection of RRM2 and NSE has the highest efficacy in diagnosing non-small cell lung cancer (NSCLC) (Area under the ROC curve (AUC) = 0.868, sensitivity = 70.9%, specificity = 90.0%).

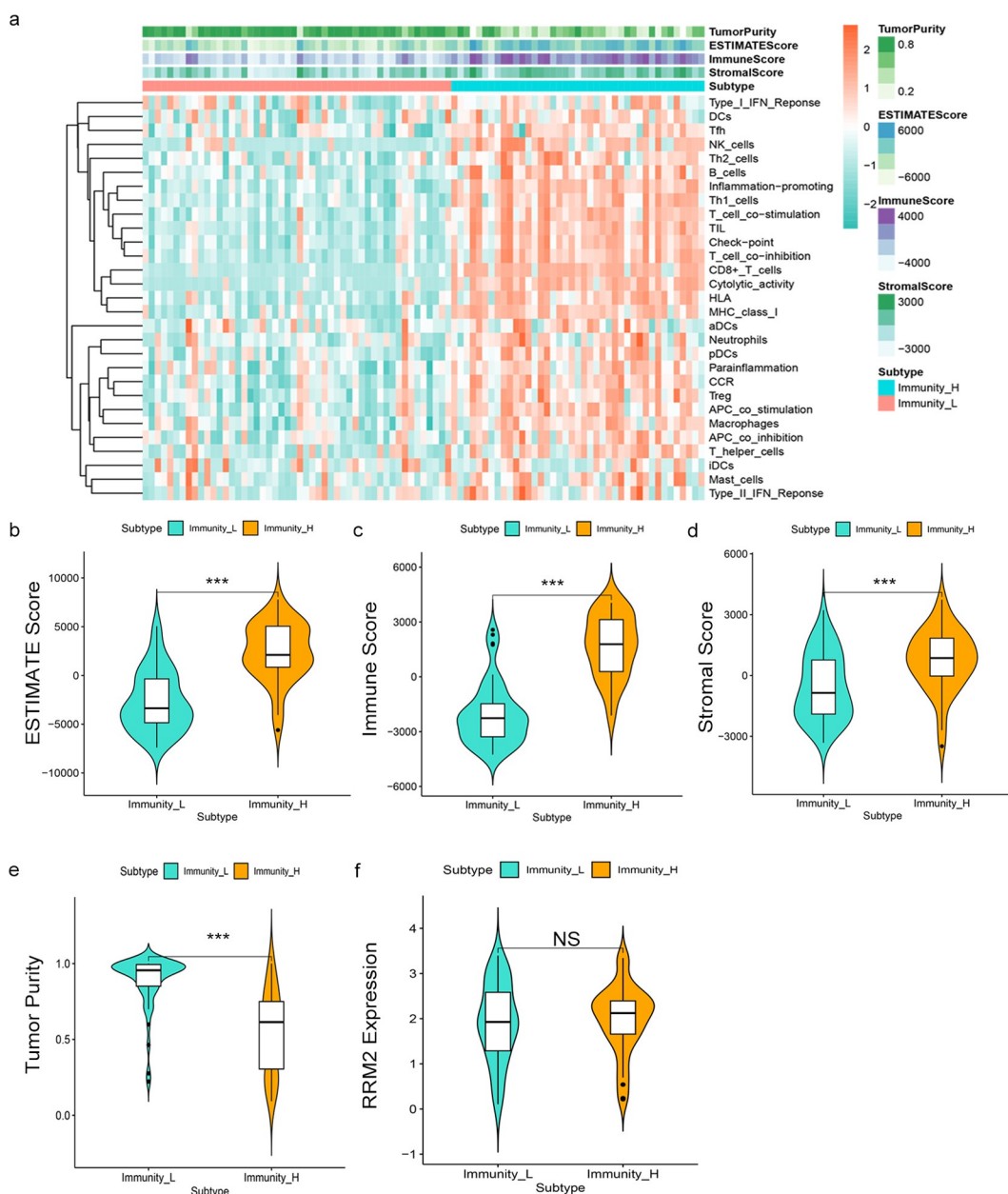

**Fig 6. Evaluation of the tumor microenvironment (TME).** (a) Heatmap of ESTIMATE scores, immune cells, stromal cells and tumor purity in high and low immune activity groups. (b-e) Relationship between ESTIMATE scores, immune cell, stromal cell, and tumor purity with immune activity in the TME. (f) Relationship between RRM2 expression and immune activity.

## Discussion

In this study, we aimed to identify potential early serum diagnostic markers for NSCLC. To do so, we first performed bioinformatics analysis of the series matrix files obtained from GSE18842, GSE19188, GSE30219, and GSE40791 using the GEO, GEPIA, and STRING databases, R, and Cytoscape. From this analysis, we identified 258 differentially expressed genes, of which we validated 10 hub genes: RRM2, CDK1, UBE2C, MAD2L1, BUB1B, CCNA2, KIF20A, BUB1, KIF11, and CCNB2. From the expression and survival prognosis analysis, we

believe that RRM2 may play a critical role in the tumorigenesis and survival prognosis of NSCLC. Then, we validated the RRM2 mRNA levels in NSCLC tissues by RT-PCR, finding significantly higher levels in both LUSC tissues and LUAD tissues than in adjacent tissues. Next, we measured RRM2 serum levels in NSCLC patients and healthy controls by ELISA, while the serum levels of CEA, CYFRA21-1, NSE, and ProGRP were analyzed by ECLIAs. Our analysis showed that RRM2 was not only significantly upregulated in NSCLC patients compared with healthy controls, but also that the serum RRM2 levels in NSCLC patients were associated with distant metastasis and histological type, but not with tumor size or lymph node metastasis. Finally, ROC curve analysis showed that the combined detection of RRM2 and NSE had the highest efficacy in diagnosing NSCLC. From the above analysis, RRM2 may serve as a potential serum diagnostic marker for NSCLC.

Recently discovered novel tumor markers have demonstrated substantial advantages over traditional ones in diagnosing NSCLC. ctDNA is a freely circulating DNA derived from apoptotic and necrotic tumor cells, present in body fluids including cerebrospinal fluid and blood, and shares high homology with tumor cell DNA. ctDNA plays a vital role in diagnosing tumor because it is minimally invasive, convenient, fast, comprehensive and reproducible in the bloodstream [31]. Although ctDNA offers numerous advantages in tumor diagnosis, its concentration in blood is significantly low, requiring the development of new techniques to improve tool sensitivity and to establish a standard detection protocol before being used in clinical diagnosis. Moreover, biomolecules such as miRNAs, proteins, and lipids enclosed within exosomes have shown significant diagnostic value in NSCLC testing [32]. Exosomes are a target of liquid biopsy, which is an easy, low-cost, and non-invasive technique. However, the usage of exosomes is restricted in clinical diagnosis due to poor reproducibility and lack of standard quantification, purification, and storage protocols. Non-coding RNA has also recently been proposed for NSCLC diagnosis [13–15]. However, its use in clinical diagnosis is still in the preliminary stage, and more efforts are required in the future. Our study found that RRM2 is a vital early serum diagnostic marker for NSCLC as it has higher sensitivity and specificity than traditional markers and is easier to incorporate into clinical diagnosis than novel markers.

In recent years, RRM2 has been implicated in tumor progression in many cancers, including breast cancer [33,34], ovarian endometriosis [35], retinoblastoma [36], renal cell carcinoma [37,38], oral squamous cell carcinoma [39], and prostate cancer [40]. Several studies have shown that RRM2 is linked to NSCLC. Han et al. confirmed that RRM2 expression is upregulated in cancer tissues compared to adjacent tissues in NSCLC patients [41]. Mah et al. discovered that RRM2 is a useful predictor of survival in certain subgroups of NSCLC patients by immunohistochemical analysis [20]. Grossi et al. also showed that RRM2 expression is associated with poor prognosis in patients with resected stage I-III NSCLC by studying 82 tumor tissues from radically resected NSCLC patients [42]. In addition, Jin et al. reported that RRM2 expression was significantly higher in LUAD tissues compared with adjacent non-cancerous tissues, and RRM2 could promote A549 cell proliferation and invasion [43]. Xie et al. demonstrated that miR-520a inhibits NSCLC progression by suppressing RRM2 and the Wnt signaling pathway [28]. As seen with the above reports, studies on RRM2 and NSCLC are limited to the tissue and cell levels, while no study has evaluated RRM2 at the serum level in NSCLC. Therefore, our study explored the association of RRM2 serum levels with NSCLC patient data for the first time. We observed elevated RRM2 levels in the serum of NSCLC patients. Moreover, compared with traditional tumor markers, RRM2 has higher sensitivity and specificity for the diagnosis of NSCLC, which helps to improve the diagnosis rate. However, RRM2 serum levels were higher in adenocarcinoma patients than in squamous cell carcinoma patients, which is inconsistent with gene expression data from the GEPIA database. Therefore,

the mechanism behind this observed inconsistency between protein and mRNA expression levels requires further investigation.

In addition, we analyzed the relationship between the RRM2 gene and the immune activity of the TME, finding that there was no correlation between RRM2 expression and immune activity. Therefore, we conclude that RRM2 is not involved in the promotion of NSCLC via the immune pathway. Notably, the KEGG pathway analysis of the hub genes revealed that RRM2 is primarily involved in the p53 signaling pathway (**S3 File**). The p53 tumor suppressor, like Bcl-2, is part of one of the major apoptotic signaling pathways. Rahman et al. observed that RRM2 can regulate the stabilization of the Bcl-2 protein, with RRM2 inhibition leading to increased Bcl-2 degradation. The two proteins can co-localize in NSCLC cells [44]. Combined with the above analysis, we hypothesize that RRM2 may be involved in the regulation of apoptosis through its interactions with the Bcl-2 pathway and that RRM2 expression may be positively correlated with NSCLC development. These hypotheses will be investigated in future studies.

Some studies have shown that low expression levels of RRM2 can help predict better prognosis in patients with advanced NSCLC who received platinum-based chemotherapy [45,46]. However, Xian-Jun et al. found that RMM2 expression had no effect on platinum-based chemotherapy response and clinical outcome in NSCLC patients [47]. Therefore, the effect of RRM2 expression in patients with NSCLC after platinum-based chemotherapy requires further study.

## Conclusion

In summary, we demonstrate that the serum levels of RRM2 were significantly increased in NSCLC patients. Our results suggest that RRM2 could serve as a potential molecular biomarker for NSCLC diagnosis. However, our study has several limitations. First, all patients in the validation cohort were Chinese. Second, the number of samples used for ELISA validation was rather small, and RRM2 was not detected in other diseases. Third, the exact mechanism of the role of RRM2 in the development and progression of NSCLC requires further investigation. Therefore, further investigation is necessary to evaluate using RRM2 as a molecular biomarker for NSCLC diagnosis, including increasing the sample size and scope, as well as determining the exact mechanism of the role of RRM2 in the development and progression of NSCLC.

## Supporting information

**S1 File. Co-expressed DEGs.**
(PDF)

**S2 File. The expression levels and survival analysis of the 10 hub genes.**
(PDF)

**S3 File. KEGG pathway analysis of hub genes.**
(PDF)

**S4 File. PCR raw data.**
(PDF)

**S5 File. Clinical raw data.**
(PDF)

## Acknowledgments

The manuscript (https://doi.org/10.21203/rs.3.rs-1366273/v1) is a preprint that has not been peer reviewed by a journal. At present, an updated version of the manuscript has been submitted to PLOS ONE.

## Author Contributions

**Conceptualization:** Dandan Zhou, Xiuming Zhai.

**Data curation:** Dandan Zhou, Xiuming Zhai.

**Formal analysis:** Dandan Zhou.

**Funding acquisition:** Dandan Zhou.

**Investigation:** Dandan Zhou, Xiuming Zhai, Ruixue Zhang.

**Methodology:** Dandan Zhou, Xiuming Zhai.

**Project administration:** Dandan Zhou.

**Resources:** Dandan Zhou, Xiuming Zhai, Ruixue Zhang.

**Software:** Dandan Zhou, Xiuming Zhai, Ruixue Zhang.

**Supervision:** Dandan Zhou, Xiuming Zhai.

**Validation:** Dandan Zhou, Xiuming Zhai.

**Visualization:** Dandan Zhou, Xiuming Zhai.

**Writing – original draft:** Dandan Zhou, Xiuming Zhai.

**Writing – review & editing:** Dandan Zhou, Xiuming Zhai.

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
