## [Decision Letter · Decision Letter 0]

23 May 2023

PONE-D-23-12628Ribonucleotide reductase regulatory subunit M2 (RRM2) as a potential sero-diagnostic biomarker in non-small cell lung cancerPLOS ONE

Dear Dr. Zhai,

Thank you for submitting your manuscript to PLOS ONE. After careful consideration, we feel that it has merit but does not fully meet PLOS ONE’s publication criteria as it currently stands. Therefore, we invite you to submit a revised version of the manuscript that addresses the points raised during the review process.

We look forward to receiving your revised manuscript.

Kind regards,

Ziheng Wang

Academic Editor

PLOS ONE

Journal Requirements:

   "This work was financially supported by the Xinglin Scholar Research Premotion Project of Chengdu University of TCM (Grant No.: YYZX2020069)."

Reviewers' comments:

Reviewer's Responses to Questions

**Comments to the Author**

1. Is the manuscript technically sound, and do the data support the conclusions?

Reviewer #1: Partly

Reviewer #2: Yes

2. Has the statistical analysis been performed appropriately and rigorously? 

Reviewer #1: Yes

Reviewer #2: Yes

3. Have the authors made all data underlying the findings in their manuscript fully available?

Reviewer #1: No

Reviewer #2: Yes

4. Is the manuscript presented in an intelligible fashion and written in standard English?

Reviewer #1: Yes

Reviewer #2: Yes

5. Review Comments to the Author

Reviewer #1: This study is interesting and provides a new target for non-small cell lung cancer, but it also has the following problems:

1、Please improve the language style, some sentences need to be more in line with English writing.

2、I would like to ask the author to improve on the introduction section, in which the author mentions the traditional markers and further mentions the progress and shortcomings of the new markers that are currently emerging, such as exosomes, circulating tumor DNA, non-coding RNA, etc.

3、In Materials and methods-Data resources. Due to involving human samples, please register the authors standardize clinical ethics, the Chinese clinical trial registration center - the world health organization (who) international clinical trial register platform registered institution ", the website: https://www.chictr.org.cn/.

4、"|log2 fold change (FC)| ≥ 2" value is 2, why not set to 1.

5、This paper mainly studies the markers of non-small cell lung cancer, which includes lung adenocarcinoma, lung squamous cell carcinoma and large cell carcinoma. The authors conclude that RRM2, UBE2C, MAD2L1, KIF20A, CDK1, and CCNB2 were statistically significant (P < 0.05). Are these genes also expressed in large cell carcinoma in the same way?

6、Please check the Data before and after. In the "Data resources" section, the blood sample is 100 examples. Why is it changed to 110 examples in the "Patient sample analysis." section?

7、Why does the "Validation and survival analysis of hub genes" part only analyze overall survival analysis, Without analyzing disease-free survival and Disease-specific survival?

8、"Tissue analysis:" this part of why the emergence of TNBC, study is not non-small cell lung cancer?

9、Please improve the discussion of this part, refer to the introduction.

Reviewer #2: The early diagnose of LUAD is a difficult problem in clinical practice. This study suggested that the single use of RRM2 has a higher diagnostic ratio for NSCLC compared with other traditional tumor markers. RRM2 is a potential sero-diagnostic biomarker for NSCLC. There are some problems in this study:

1. It is suggested to add some introduction about RRM2 in the introduction section.

2. It is suggested to draw a volcano map and mark the location of RRM2 gene to describe the DEGs more vividly.

3. The bioinformatics analysis of RRM2 function is less, it is suggested to add the bioinformatics analysis such as immunity or drug resistance correlation.

4. It is suggested that the up-regulated genes and down-regulated genes should be analyzed by GO/KEGG separately.

5. English writing needs further improvement，It is suggested that article should be revised by native English speakers.

In general, there are some problems in this manuscript. Manuscript should be carefully reviewed and revised by the author before publication.

6. PLOS authors have the option to publish the peer review history of their article (what does this mean?). If published, this will include your full peer review and any attached files.

Reviewer #1: **Yes: **Kangle Zhu

Reviewer #2: No

---

## [Author Response · Author response to Decision Letter 0]

9 Jun 2023

Response to Reviewer #1:

1. Please improve the language style, some sentences need to be more in line with English writing.

Response: Thank you for this comment. We have invited Editage (www.editage.com) for English language editing. 

2. I would like to ask the author to improve on the introduction section, in which the author mentions the traditional markers and further mentions the progress and shortcomings of the new markers that are currently emerging, such as exosomes, circulating tumor DNA, non-coding RNA, etc.

Response: Thank you for your constructive suggestion. As you suggested, the relevant content has been added to the introduction section. The additions are as follows: Notably, several novel biomarkers for diagnosis of NSCLC have been identified in recent years due to their higher sensitivity and specificity compared to traditional tumor markers, including exosomes[9, 10], circulating tumor DNA (ctDNA)[11, 12], microRNAs[13], CircRNAs[14],and long non-coding RNA[15, 16]. Nonetheless, there are still many challenges associated with their widespread use in population screening, including restrictive detection technology, equipment complexity, high testing costs.

3. In Materials and methods-Data resources. Due to involving human samples, please register the authors standardize clinical ethics, the Chinese clinical trial registration center - the world health organization (who) international clinical trial register platform registered institution ", the website: https://www.chictr.org.cn/.

Response: Thank you for this comment. We have made a retrospective registration in "the Chinese clinical trial registration center - the world health organization (who) international clinical trial register platform registered institution", but as of now the status is still “Not be verified”.

4. "|log2 fold change (FC)| ≥ 2" value is 2, why not set to 1.

Response: Thank you for your consideration. Indeed, we have tried different fold change values. If |log2 fold change (FC)| ≥ 1, there are too many differential genes and we can't screen key genes accurately and effectively. The |log2 fold change (FC)| ≥ 2 is the most appropriate value that determined after constant trials.

5. This paper mainly studies the markers of non-small cell lung cancer, which includes lung adenocarcinoma, lung squamous cell carcinoma and large cell carcinoma. The authors conclude that RRM2, UBE2C, MAD2L1, KIF20A, CDK1, and CCNB2 were statistically significant (P < 0.05). Are these genes also expressed in large cell carcinoma in the same way?

Response: Thank you very much for carefully reviewing of our manuscript. It is very helpful for our improvement. We further validated the expression levels of the 10 hub genes in NSCLC and normal tissues using the GEPIA database, but only LUAD and LUSC were included in the GEPIA database. Following your suggestion, we reviewed the available literature for the expression of RRM2, UBE2C, MAD2L1, KIF20A, CDK1, and CCNB2 reported in large cell lung carcinoma. Now, we have added it in the revised manuscript. The additions are as follows:

Based on the available reports, we found that the mRNA and protein levels of RRM2[28], UBE2C[29], and CDK1[30] were significantly upregulated in the large cell lung cancer cell line NCI-H460, whereas the expression of MAD2L1, KIF20A, and CCNB2 in the large cell lung cancer cell line NCI-H460 has not been reported.

6. Please check the Data before and after. In the "Data resources" section, the blood sample is 100 examples. Why is it changed to 110 examples in the "Patient sample analysis." section?

Response: Thank you very much for patiently reviewing of our manuscript. It is our carelessness. The blood sample in the non-small cell lung cancer group is 110 examples. We have corrected this in the "Data resources" section.

7. Why does the "Validation and survival analysis of hub genes" part only analyze overall survival analysis, without analyzing disease-free survival and Disease-specific survival?

Response: Thank you for your consideration. This is very helpful for us to improve. Following your suggestion, we have added RFS analysis, as the GEPIA database currently only contains overall survival and disease-free survival (DFS) analysis. The results of the RFS analysis have been added to Supplementary file 2. The additions in revised manuscript are as follows: RFS results showed that high expression of RRM2, KIF20A, CDK1, CCNB2, CCNA2, BUB1B and BUB1 in LUAD patients were associated with shorter survival in LUAD patients (P < 0.05), whereas expression levels of UBE2C, MAD2L1 and KIF11 were not associated with survival in LUAD patients (P > 0.05). RFS in LUSC patients was not associated with the expression levels of any of the 10 hub genes (P > 0.05).

8. "Tissue analysis:" this part of why the emergence of TNBC, study is not non-small cell lung cancer?

Response: Thank you for your careful reviewing of our manuscript. I'm very sorry for our carelessness. This is a writing error. We have corrected it in the revised manuscript. "TNBC" has been corrected to "NSCLC".

9. Please improve the discussion of this part, refer to the introduction.

Response: Thank you for your constructive comments. As you suggested, we have improved the relevant content in the discussion section. The additions are as follows: Recently discovered novel tumor markers have demonstrated substantial advantages over traditional ones in diagnosing NSCLC. ctDNA is a freely circulating DNA derived from apoptotic and necrotic tumor cells, present in body fluids including cerebrospinal fluid and blood, and shares high homology with tumor cell DNA. ctDNA plays a vital role in diagnosing tumor because it is minimally invasive, convenient, fast, comprehensive and reproducible in the bloodstream[31]. Although ctDNA offers numerous advantages in tumor diagnosis, its concentration in blood is significantly low, requiring the development of new techniques to improve tool sensitivity and to establish a standard detection protocol before being used in clinical diagnosis. Moreover, biomolecules such as miRNAs, proteins, and lipids enclosed within exosomes have shown significant diagnostic value in NSCLC testing[32]. Exosomes are a target of liquid biopsy, which is an easy, low-cost, and non-invasive technique. However, the usage of exosomes is restricted in clinical diagnosis due to poor reproducibility and lack of standard quantification, purification, and storage protocols. Non-coding RNA has also been proposed for NSCLC diagnosis recently[13-15]. However, its use in clinical diagnostics is still in the preliminary phase, and more efforts are required in the future. Our study found that RRM2 is a vital early serum diagnostic marker for NSCLC as it has higher sensitivity and specificity than traditional markers and is easier to incorporate into clinical diagnosis than novel markers.

Response to Reviewer #2:

1. It is suggested to add some introduction about RRM2 in the introduction section.

Response: Thank you very much for carefully reviewing of our manuscript. This is very helpful for us to improve. As you suggested, we have added it in the introduction section. The additions are as follows: Ribonucleotide reductase regulatory subunit M2 (RRM2) encodes one of the two different subunits of ribonucleotide reductase, which catalyzes the conversion of ribonucleotides to deoxyribonucleotides[19]. Several studies have reported increased expression of RRM2 mRNA in NSCLC tissues[20, 21] and cell lines[16, 21]. However, it is unclear whether RRM2 can be used as a diagnostic marker for NSCLC. RRM2 protein is a secreted protein and can be detected in serum. In this study, we aim to investigate the possibility of RRM2 as a serum diagnostic marker for NSCLC based on bioinformatics.

2. It is suggested to draw a volcano map and mark the location of RRM2 gene to describe the DEGs more vividly.

Response: Thanks for your valuable suggestions. As you suggested, we have added volcano maps for each of the datasets in Figure 1 and marked the location of the RRM2 gene on each volcano map.

Fig. 1 Data from GSE18842, GSE19188, GSE30219 and GSE40791 were used to identify differentially expressed genes (DEGs) between non-small cell lung cancer (NSCLC) and normal samples. The volcano plots show all the expressed genes from GSE18842 (a), GSE19188 (b), GSE30219 (c), GSE40791 (d). Red color represents up-regulated genes, mediumaquamarine color represents down-regulated genes. (e) A total of 258 co-expressed genes were screened. (f) 91 genes were significantly up-regulated and (g) 167 genes were significantly down-regulated.

3. The bioinformatics analysis of RRM2 function is less, it is suggested to add the bioinformatics analysis such as immunity or drug resistance correlation.

Response: Thank you for your constructive comments. As you suggested, we have added the analysis of the relationship between RRM2 gene and immune activity in the tumour microenvironment. 

The additions in “Materials and methods” are as follows: 

The relevance of hub genes to immune activity in the tumour microenvironment (TME).First, the single sample gene set enrichment analysis (ssGSEA) algorithm was used to normalize the gene expression values of the NSCLC samples. Then, we applied the empirical cumulative distribution function[25] to calculate the enrichment fraction of 29 immune cell types[26], which facilitated the categorization of NSCLC samples into distinct immune activity sets. Subsequently, we employed the GSVA software package to stratify patients into high and low immune activity groups. To validate the accuracy of immune activity grouping based on the GSE19188 data, we utilized the ESTIMATE method[27], which reflects the infiltration levels of immune cells and stromal cells within TME, based on their specific gene expression levels.

The additions in “Results” are as follows:

The relationship between RRM2 gene and immune activity in TME. In addition, we further assessed the relationship between RRM2 and immune activity to elucidate the mechanism by which RRM2 promotes the NSCLC development. Firstly, we evaluated the GSE19188 expression matrix using the ESTIMATE algorithm to obtain the immune cell score, stromal cell score, ESTIMATE score (total score), and tumor purity score in TME (Figure 6a). Further analysis demonstrated that the ESTIMATE score, immune cell score, and stromal cell score were positively correlated with immune activity (Figure 6b-6d), while tumor purity showed an inverse correlation with immune activity (Figure 6e). We ultimately found no correlation between RRM2 expression and immune activity (P＞0.05) in Figure 6f.

Fig. 6 Evaluation of the tumor microenvironment (TME). (a) Heatmap of ESTIMATE scores, immune cells, stromal cells and tumor purity in high and low immune activity groups. (b-e) Relationship between ESTIMATE scores, immune cell, stromal cell, and tumor purity with immune activity in the TME. (f) Relationship between RRM2 expression and immune activity.

The additions in “Discussion” are as follows:

In addition, we analyzed the relationship between the RRM2 gene and the immune activity of the TME, finding that there was no correlation between RRM2 expression and immune activity.

4. It is suggested that the up-regulated genes and down-regulated genes should be analyzed by GO/KEGG separately.

Response: Thank you for your constructive suggestion. It is very helpful for our improvement. As you suggested, we have analyzed the up-regulated and down-regulated DEGs by GO/KEGG separately (Figure 2).

Fig. 2 The biological function of the 258 differentially expressed genes (DEGs). GO and KEGG enrichment analysis of up-regulated DEGs (a) and down-regulated DEGs (b).

5. English writing needs further improvement，It is suggested that article should be revised by native English speakers.

Response: Thank you for this comment. We have invited Editage (www.editage.com) for English language editing.

---

## [Decision Letter · Decision Letter 1]

18 Jul 2023

PONE-D-23-12628R1Ribonucleotide reductase regulatory subunit M2 (RRM2) as a potential sero-diagnostic biomarker in non-small cell lung cancerPLOS ONE

Dear Dr. Zhai,

Thank you for submitting your manuscript to PLOS ONE. After careful consideration, we feel that it has merit but does not fully meet PLOS ONE’s publication criteria as it currently stands. Therefore, we invite you to submit a revised version of the manuscript that addresses the points raised during the review process.

We look forward to receiving your revised manuscript.

Kind regards,

Ziheng Wang

Academic Editor

PLOS ONE

Journal Requirements:

Additional Editor Comments:

Please revise the manuscript as the reviewers' requirements.

Reviewers' comments:

Reviewer's Responses to Questions

**Comments to the Author**

1. If the authors have adequately addressed your comments raised in a previous round of review and you feel that this manuscript is now acceptable for publication, you may indicate that here to bypass the “Comments to the Author” section, enter your conflict of interest statement in the “Confidential to Editor” section, and submit your "Accept" recommendation.

Reviewer #1: All comments have been addressed

Reviewer #2: (No Response)

2. Is the manuscript technically sound, and do the data support the conclusions?

Reviewer #1: Yes

Reviewer #2: (No Response)

3. Has the statistical analysis been performed appropriately and rigorously? 

Reviewer #1: Yes

Reviewer #2: (No Response)

4. Have the authors made all data underlying the findings in their manuscript fully available?

Reviewer #1: Yes

Reviewer #2: (No Response)

5. Is the manuscript presented in an intelligible fashion and written in standard English?

Reviewer #1: No

Reviewer #2: (No Response)

6. Review Comments to the Author

Reviewer #1: Thank you very much for your revision. Now I still have the following questions:

1、Why is the annotation not modified in the manuscript, and the color is not prominent in the modified area? I don't know what specific content you modified?

2、EVOlyzer-2 150 platform. What is this?

Reviewer #2: (No Response)

7. PLOS authors have the option to publish the peer review history of their article (what does this mean?). If published, this will include your full peer review and any attached files.

Reviewer #1: No

Reviewer #2: No

---

## [Author Response · Author response to Decision Letter 1]

18 Aug 2023

Response to Reviewer #1:

1. Why is the annotation not modified in the manuscript, and the color is not prominent in the modified area? I don't know what specific content you modified?

Response: Thank you for your careful and patient consideration of my manuscript. Please allow me to explain your confusion.

(1) I can only see your comments in the PLOS ONE submission system, but I do not have access to your annotated manuscript. After communicating with the PLOS ONE editor several times, I have still not been able to access your annotated manuscript. However, we have carefully revised the linguistic aspects of the revised manuscript again, based on your latest suggestions. 

(2) As the PLOS ONE submission system requires the author to upload "Revised Manuscript with Track Changes.docx" and "Manuscript -ClearCopy.docx", I can't mark up the changes again in the "Manuscript -ClearCopy.docx". I've now marked this revision in red and underlined the previous revision in "Revised Manuscript with Track Changes.docx".

Thank you again for your valuable time in helping our team improve academically. 

2. EVOlyzer-2 150 platform. What is this?

Response: Thank you for your consideration. The Freedom EVOlyzer is a fully automated enzyme immunoassay instrument and the 150 is a model of this instrument.

---

## [Editor Report · Decision Letter 2]

30 Aug 2023

Ribonucleotide reductase regulatory subunit M2 (RRM2) as a potential sero-diagnostic biomarker in non-small cell lung cancer

PONE-D-23-12628R2

Dear Dr. Zhai,

We’re pleased to inform you that your manuscript has been judged scientifically suitable for publication and will be formally accepted for publication once it meets all outstanding technical requirements.

Kind regards,

Ziheng Wang

Academic Editor

PLOS ONE
---

## [Editor Report · Acceptance letter]

1 Sep 2023

PONE-D-23-12628R2 

Ribonucleotide reductase regulatory subunit M2 (RRM2) as a potential sero-diagnostic biomarker in non-small cell lung cancer 

Dear Dr. Zhai:

I'm pleased to inform you that your manuscript has been deemed suitable for publication in PLOS ONE. Congratulations! Your manuscript is now with our production department. 

Kind regards, 

on behalf of

Professor Ziheng Wang 

Academic Editor

PLOS ONE